# Novel p38 Mitogen-Activated Protein Kinase Inhibitor Reverses Hypoxia-Induced Pulmonary Arterial Hypertension in Rats

**DOI:** 10.3390/ph15070900

**Published:** 2022-07-21

**Authors:** Grazielle Fernandes Silva, Jaqueline Soares da Silva, Allan Kardec Nogueira de Alencar, Marina de Moraes Carvalho da Silva, Tadeu Lima Montagnoli, Bruna de Souza Rocha, Rosana Helena Coimbra Nogueira de Freitas, Roberto Takashi Sudo, Carlos Alberto Manssour Fraga, Gisele Zapata-Sudo

**Affiliations:** 1Programa de Pesquisa em Desenvolvimento de Fármacos, Instituto de Ciências Biomédicas, Universidade Federal do Rio de Janeiro, Rio de Janeiro 21941-902, RJ, Brazil; graziellemed@gmail.com (G.F.S.); ssjck@hotmail.com (J.S.d.S.); allankdc@gmail.com (A.K.N.d.A.); marina-mcs@hotmail.com (M.d.M.C.d.S.); tmontagnoli@gmail.com (T.L.M.); brunadesouzarocha.98@gmail.com (B.d.S.R.); freitasrh@yahoo.com.br (R.H.C.N.d.F.); or rtsudo@icb.br (R.T.S.); 2Programa de Pós-Graduação em Cardiologia, Instituto do Coração Edson Saad, Universidade Federal do Rio de Janeiro, Rio de Janeiro 21941-913, RJ, Brazil; 3Programa de Pós-Graduação em Farmacologia e Química Medicinal, Instituto de Ciências Biomédicas, Universidade Federal do Rio de Janeiro, Rio de Janeiro 21941-902, RJ, Brazil

**Keywords:** pulmonary hypertension, right ventricle failure, MAPK inhibitor, SU5416/hypoxia, inflammation

## Abstract

Mitogen-activated protein kinase (MAPK) signaling is strongly implicated in cardiovascular remodeling in pulmonary hypertension (PH) and right ventricle (RV) failure. The effects of a newly designed p38 inhibitor, LASSBio-1824, were investigated in experimentally induced PH. Male Wistar rats were exposed to hypoxia and SU5416 (SuHx), and normoxic rats were used as controls. Oral treatment was performed for 14 days with either vehicle or LASSBio-1824 (50 mg/kg). Pulmonary vascular resistance and RV structure and function were assessed by echocardiography and catheterization. Histological, immunohistochemical and Western blot analysis of lung and RV were performed to investigate cardiovascular remodeling and inflammation. Treatment with LASSBio-1824 normalized vascular resistance by attenuating vessel muscularization and endothelial dysfunction. In the heart, treatment decreased RV systolic pressure, hypertrophy and collagen content, improving cardiac function. Protein content of TNF-α, iNOS, phosphorylated p38 and caspase-3 were reduced both in lung vessels and RV tissues after treatment and a reduced activation of transcription factor c-fos was found in cardiomyocytes of treated SuHx rats. Therefore, LASSBio-1824 represents a potential candidate for remodeling-targeted treatment of PH.

## 1. Introduction

Pulmonary hypertension (PH) is a clinical condition in which remodeling and deleterious obstruction of the pulmonary vessels are observed and culminate in a significant rise in pulmonary arterial resistance and pressure [1]. It affects approximately 10% of the elderly population worldwide [2] and 3–10 new cases per million are diagnosed each year, mostly in young women [3]. When no adequate treatment is used, a mean survival of 2.8 years is expected, and, even considering recent advances in therapeutics, the survival rate remains only 61% after 5 years of diagnosis [3,4].

Low survival rates are associated with right-ventricle (RV) dysfunction and failure, as consequences of increased pulmonary-vessel tone and vascular remodeling [1,5,6]. Currently available treatment for PH targets the reduction of pulmonary arterial vasoconstriction but does not significantly attenuate vascular remodeling or pathophysiological mechanisms involved in the advent of heart failure [1,5]. Alternative targets are highlighted in studies of drug development, which could delay the rapid progression of vascular lesions and RV dysfunction.

The events involved in cardiac and vascular remodeling observed in PH are conducted by abnormal smooth-muscle cells, endothelium, fibroblasts, myofibroblasts and perivascular immune cells [1,6,7,8,9,10,11]. Therefore, it is important to consider the multiple molecular signaling pathways in order to explore novel therapeutic targets for PH treatment [5,9,12]. Recently, sotatercept, which counterbalances hypertrophic stimuli caused by decreased signaling through bone morphogenetic protein receptor 2, is ultimately reducing pulmonary vascular resistance and cardiac hypertrophy [13].

Mitogen-activated protein kinases (MAPK) trigger pathophysiological mechanisms in several diseases, by modulating cell proliferation, differentiation, survival and tissue inflammation [14,15,16,17,18]. Inhibition of p38 has shown beneficial effects in animal models of both PH and RV failure [9,12,19,20,21]. Activation of p38 is implicated in increased production of inflammatory mediators, such as tumor necrosis factor (TNF)-α and inducible nitric oxide synthase (iNOS) [22,23,24]. In addition, p38 also regulates cell apoptosis mediated by caspases [14,15,16], which is an important condition involved in vascular remodeling during PH. Moreover, an independent role in RV hypertrophy and dysfunction in response to pressure overload was also demonstrated for this MAPK [14,19]. Thus, p38 inhibition in the cardiopulmonary system could promote beneficial effects directly in both RV and lungs with slight systemic side effects.

Recently, the synthesis of (*E*)-*N*’-(4-(pyridin-2-yl)benzylidene)-2-naphthohydrazide (LASSBio-1824, Figure 1) was reported [25]. This new *N*-acylhydrazone derivative inhibited p38 in an in vitro biochemical assay and showed anti-inflammatory activity in vivo after oral administration in mice [25]. Based on this, the present work investigated the pharmacological effects of LASSBio-1824 in the pathophysiological changes in structure and function of the cardio-pulmonary system in rats with hypoxia+SU5416-induced (SuHx) PH.

## 2. Results

### 2.1. LASSBio-1824 Selectively Decreases Pulmonary Vascular Resistance

Increased vascular resistance was observed after induction of PH due to a significant reduction of the ratio of pulmonary flux acceleration time (PAT) to total ejection time (PET) detected on Doppler transthoracic echocardiography (TTE) in rats submitted to SuHx model (*p* < 0.05; Figure 2A). When orally administered for 2 weeks, LASSBio-1824 significantly reduced vascular resistance in pulmonary vascular bed, as indicated by increased PAT/PET ratio when compared with vehicle-treated SuHx rats (*p* < 0.05; Figure 2A). At the end of protocol, the appearance of a mid-systolic notch was evident in the flow profile in pulmonary arteries from SuHx + vehicle group, while LASSBio-1824 normalized the flow shape (Figure 2B upper). Meanwhile, no alterations in body weight or systemic hemodynamics were observed between SuHx and normoxic groups, regardless of treatment (Table 1).

### 2.2. LASSBio-1824 Reduces Medial Hypertrophy and Endothelial Dysfunction in Pulmonary Arteries

After five weeks of protocol, smooth-muscle actin immunostaining of lung sections showed that SuHx + vehicle rats displayed a significant thickening of distal pulmonary vessels when compared to normoxic rats (*p* < 0.05; Figure 2B middle, and Figure 2C). Medial hypertrophy was accompanied by a marked expansion of perivascular collagen, as noticed in picro-Sirius red stain (*p* < 0.05; Figure 2B lower, and Figure 2D). Such changes were not observed in SuHx + LASSBio-1824 group (*p* < 0.05; Figure 2C,D), indicating that oral treatment for 14 days with this *N*-acylhydrazone significantly reduced the proliferation of smooth-muscle cells and fibroblasts in vascular walls.

Endothelial-mediated vasodilatation of pulmonary arteries was impaired in SuHx + vehicle animals (*p* < 0.05; Figure 2E and Table 1), as indicated by a reduction in acetylcholine (ACh)-induced maximal relaxation. In contrast, endothelial function was preserved in SuHx rats treated with LASSBio-1824, as indicated by increased vasodilatory response to ACh compared to SuHx + vehicle rats (*p* < 0.05). No changes were observed in contractile responses of pulmonary arteries to phenylephrine (Table 2). In addition, both medial hypertrophy and endothelial function displayed significant correlations with the improvement in pulmonary vascular resistance (*p* < 0.05; Figure 2F,G).

### 2.3. LASSBio-1824 Alleviates RV Overload and Remodeling and Preserves RV Function

While no significant differences were detected in LV hemodynamics and structure between groups (Table 3), increased pulmonary vascular resistance in the SuHx + vehicle group resulted in elevated RV systolic pressures and severe reduced RV cardiac output (*p* < 0.05; Figure 3A and Table 3). SuHx-induced RV overload led to a significant ventricular hypertrophy, as indicated by both TTE (Figure 3B upper and Figure 3C) and Fulton index (Figure 3D). Moreover, RV dilatation was also detected in echocardiographic analysis in SuHx rats (Figure 3B lower and Figure 3E). Reduction in vascular resistance by oral treatment with LASSBio-1824 resulted in normalization of RV systolic pressure and cardiac function, as well as preservation of RV thickness and area in rats with PH (*p* < 0.05; Figure 3A–E and Table 3).

A similar trend was observed in histologic analysis of RV tissues, since increased interstitial cell numbers and collagen deposition were found in SuHx rats and LASSBio-1824 significantly reduced both parameters (*p* < 0.05; Figure 4A–C).

### 2.4. Administration of LASSBio-1824 Normalized Inflammation in Lung and RV

When compared to the normoxia group, lung immunohistochemistry in SuHx + vehicle rats revealed a significant increase in vascular protein content of important inflammatory mediators, such as TNF-α and iNOS, which resembled p38 MAPK expression (*p* < 0.05; Figure 5A–D). Oral treatment with LASSBio-1824 greatly attenuated vessel inflammatory marker content and p38 protein expression (*p* < 0.05; Figure 5A–D).

Furthermore, the increase in TNF-α and iNOS was reinforced by using Western blot analysis, and exacerbated activation of p38 and caspase 3 was observed in RV homogenates from vehicle-treated SuHx animals (*p* < 0.05; Figure 6A–B). LASSBio-1824 normalized the expression of inflammatory markers and caspase activation, matching the reduction seen in p38 phosphorylation (Figure 6A–B). Immunohistochemical staining displayed intense nuclear localization of c-fos in RV from SuHx + vehicle rats, possibly in response to higher cardiomyocyte stress (*p* < 0.05; Figure 6C,D) which was attenuated by LASSBio-1824. The reduction in the activity of this transcription factor in RV myocytes indicates possible inhibition of the early gene program and consequent cell dysfunction and apoptosis (*p* < 0.05; Figure 6C,D).

## 3. Discussion

Although the number of therapies for PH has increased in the last years, currently used vasodilators do not directly address the cardiovascular remodeling associated with disease progression. Therefore, the investigation of new targets that regulate tissue remodeling is of great interest for the development of new drugs for treatment of PH. Recently, the importance of inflammation to the pathways involved in tissue remodeling has gained attention [8,10], and drugs with anti-inflammatory activity represent a promising alternative for management of disease progression.

PH is a multifactorial disease associated with low survival rate in the absence of adequate treatment [1,3,4]. Mortality is correlated to RV dysfunction, which in turn is a consequence of increased pulmonary vascular resistance and remodeling [1,5,6]. Many cell types and stimuli are involved in the initiation, evolution and stabilization of vascular lesions [6,7,8,9,10,11]. Inflammatory stimuli in the vessel wall not only induces infiltration of circulating leukocytes but also stimulates endothelial and smooth-muscle cells to oppose apoptosis, resulting in a proliferative condition [8,10]. Although these aspects may involve different intracellular pathways, some enzymes such as MAPK are important key factors [12,14,15,16,17,18]. Among MAPK enzymes, the importance of p38 in PH was previously reported, because its expression and activity are increased in patients [12,20,21,26]. The relevance of p38 activation to PH pathogenesis is evidenced by its involvement in different cellular processes triggered by growth factors, inflammatory cytokines, reactive oxygen species, hypoxia and environmental stress [14,15,18]. Signaling through p38 interferes with the balance between cell survival and apoptosis, and mediates inflammatory cytokine production locally [14,15,16,17,18]. Recently, a noncanonical pathway was demonstrated which may also contribute to MAPK signaling perpetuation by a positive feedback of p38 activation [14,15].

When activated, p38 translocates from cytoplasm to nucleus and phosphorylate transcription factors involved in inflammation, cell proliferation and apoptosis [14,15]. Increased p38 expression and activity found in lung vessels from patients with idiopathic PH further suggest a key role for this MAPK in pulmonary vascular remodeling [20,21,26], in addition to a selective pulmonary vasoconstriction in response to hypoxia [26]. Those features could significantly account for the vascular remodeling in the small circulation [11], hypertrophy and dysfunction of RV with subsequent failure. Furthermore, the role of p38 in cardiac and vascular remodeling was previously described in rodent models of PH and pressure-overload-induced RV dysfunction [19,20,21].

*N*-acylhydrazones are known for their extensive beneficial biological activities, including anti-inflammatory effects [27,28]. LASSBio-1824 is a new N-acylhydrazone that demonstrated p38 inhibiting activity in vitro and promoted a reduction in inflammatory cell migration by attenuating cytokine production in mice [25]. Oral administration of this compound displayed improved anti-TNF activity when compared to SB203580, suggesting a better drug-likeness profile than currently studied p38 inhibitors [25]. Therefore, the kinase inhibition activity and anti-inflammatory effects in vivo prompted the study of LASSBio-1824 as a potential treatment for PH.

PH is marked by an increase in pulmonary vascular resistance caused by increased tone and remodeling in intrapulmonary arterioles. The SuHx model exhibits several characteristics of human PH, including histological and hemodynamic alterations [29,30,31]. Herein, it was demonstrated that SuHx rats developed PH after 21 days due to slower blood flow through the pulmonary arteries, as indicated by short PAT/PET ratio. Moreover, elevated pulmonary vascular resistance and vessel stiffness lead to premature wave reflection towards the RV during systole, impairing RV ejection and causing a notch in the Doppler profile [32,33], an alteration seen in late-stage PH [33,34]. Treatment with LASSBio-1824 normalized the pulmonary blood-flow wave shape and abolished the mid-systolic notch, which accompanied the increase in PAT/PET, indicating a normalization of blood flow through pulmonary circulation.

Increased vascular tone on pulmonary arteries results from an association of increased vessel stiffness and decreased endothelial-mediated vasodilation. Decreased endothelial function results of impaired production of vasodilators [9,11], contributing to increased pulmonary vascular resistance, suggesting a significant correlation between PAT/PET ratios and ACh-induced maximal relaxation in SuHx rats. Endothelial dysfunction occurs in PH due to endothelial phenotype changes associated with excessive release of growth factors, chemokines, cytokines and vasoconstrictors [1]. The actions of these mediators converge to p38 activation, resulting in exacerbated cell proliferation, resistance to apoptosis and inflammation observed in PH subjects [20,35,36,37]. LASSBio-1824 enhanced ACh response in pulmonary arteries from SuHx rats, normalizing the endothelial control of vascular tone, suggesting that p38 inhibition decreased endothelial phenotype transition.

Pulmonary vascular remodeling contributes not only to raise vascular tone but also to reduce vessel wall compliance. Increased wall thickness and perivascular fibrosis are among the most common alterations seen in vascular lesions from PH patients [1]. In this study, increased muscularization and perivascular collagen content were found in lung arterioles from SuHx + vehicle animals, indicating significant vessel remodeling in this group. This alteration affected the pulmonary vascular resistance, as PAT/PET ratios were negatively correlated to medial wall area.

Vascular remodeling results from activation of smooth-muscle cells and fibroblasts [1,38], which involve p38 MAPK pathways [22,26]. Increased p38 content is found in lung arterioles and peripheral blood leukocytes from PH patients [20,39], and its activation upregulates TNF-α and iNOS in the vessel wall, stimulating proliferation, arteriole muscularization and collagen synthesis [16,22,26,40,41]. In addition, TNF-α secretion in the vasculature also promotes endothelial permeability, facilitating the infiltration of circulating leukocytes and aggravating tissue inflammation [42,43,44]. Our data demonstrate not only an increase in iNOS and TNF-α expression, but also higher p38 content in pulmonary vessels of SuHx rats, which indicates abnormal p38 MAPK signaling in these remodeled arterioles. The effect of LASSBio-1824 on vascular remodeling is concomitant to the normalization in p38 protein content, presumably by limiting the signaling through this MAPK and reducing TNF-α and iNOS concentration in arterioles.

The persistent afterload induced by PH promotes adaptive changes on RV structure, leading to dysfunction and failure [12,19]. Increased RV systolic pressure was found in SuHx rats, which affected RV structure and function, as seen by its hypertrophy, dilatation and impaired output. However, after LASSBio-1824 treatment those parameters were similar to normoxic controls, which could be due, in part, to the attenuation in vascular remodeling and consequent afterload, since p38 MAPK signaling is also directly implicated in ventricular remodeling [19].

Increased protein content and phosphorylation are observed in RV in PH models, where it stimulates cardiac fibroblasts through myocardin-related transcription factor and metalloproteinase-9 activation [19,45]. Elevated interstitial cell density and collagen content were seen in RV from SuHx rats, indicating increased stromal cell proliferation and activation in response to cardiac overload. LASSBio-1824 reversed these alterations, reducing RV fibrosis and cell density in SuHx animals. Simultaneously to these findings, a reduction in p38 phosphorylation was found, which implies noncanonical activation of this enzyme by a positive feedback loop, mediated by the scaffold protein and p38 substrate TAB1 [14,15].

p38 MAPK signaling is directly involved in the progression of ventricular dysfunction, because it promotes cardiomyocyte adaptation to different stressors, which in turn triggers mechanisms of cardiac inflammation and further stimulates fibrosis [14,15,46,47]. Cardiac activation of caspase-3 leads to an ongoing apoptotic process in RV tissue in PH rats, indicating a transitioning to failing phenotype in RV myocytes [16,48]. This finding is related to the chamber dilatation seen in TTE images in the SuHx + vehicle group. Chronic mechanical overload leads to cardiomyocyte stress and activation of early response pathways, such as increased c-fos expression and activity [49,50]. Transcription of the *FOS* gene is dependent on serum response factor activation, which in turn is a downstream target of p38 MAPK signaling [18,19]. Therefore, increased c-fos nuclear expression in cardiomyocytes indicates increased p38 phosphorylation and activity in these cells, as seen by the concomitant increase in both parameters in SuHx rats. In contrast, LASSBio-1824 reduced caspase-3 activation by reducing cardiomyocyte stress signaling, as suggested by attenuated c-fos expression in myocyte nuclei through inhibition of p38 signaling.

Cardiac expression of TNF-α and iNOS is also dependent on p38 activity [15,23,24] and has been associated with the development of RV hypertrophy and progression to RV failure [51,52,53,54,55]. Increased content of both mediators was found in the SuHx + vehicle group and was normalized by treatment with LASSBio-1824, correlating their expressions to activated p38 levels. Moreover, since those proteins are responsible for the cardiac inflammation and fibrosis [27,56,57], the effect of LASSBio-1824 on attenuating their expression also contributes to the reduction in cardiac collagen content.

## 4. Materials and Methods

### 4.1. Drugs and Reagents

LASSBio-1824 and SU5416 were provided by Laboratório de Avaliação e Síntese de Substâncias Bioativas (LASSBio^®^; Universidade Federal do Rio de Janeiro, Rio de Janeiro Brazil). Dimethylsulfoxide, ketamine and xylazine were gently provided by Cristália Produtos Químicos e Farmacêuticos (Itapira, São Paulo, Brazil). Antibodies used in this work are described in Table 4. All other reagents were purchased from Sigma-Aldrich (St. Louis, MO, USA).

### 4.2. Animals and Experimental Design

All experimental protocols were approved by the Committee on Ethics in the Use of Animals at Universidade Federal do Rio de Janeiro (license number 039/19). Male Wistar rats (180–250 g) originated from the Animal Facility of Universidade Federal do Rio de Janeiro and were housed in cages containing 3–4 animals at 21 ± 1 °C, under a light/dark cycle of 12 h with free access to chow and water. Rats were kept in accordance with the Brazilian Guide of Production, Maintenance and Utilization of Animals for Teaching or Scientific Research Activities (1st edition, 2016) approved by the National Council for Control of Animal Experimentation.

PH was induced in rats by administration of SU5416 during exposure to hypoxia, which reproduces the severe and persistent vascular remodeling and RV failure similar to late-stage human disease [29,30,58]. This rat model, SuHx, develops more severe features than hypoxia only [29,30,31] with a progressive proliferative endotheliopathy caused by SU5416 after their return to normoxia [29,30].

Figure 7 shows the delineation of the experimental protocol.

Rats were randomly divided into two groups: Normoxia (*n* = 5), who were maintained in room air and used as controls; and SuHx (*n* = 10), in which PH was induced. Animals in the SuHx group were subjected to 3 weeks of normobaric hypoxia in a ventilated acrylic chamber, where atmospheric oxygen concentration was kept at 10% by an oxycycler controller (ProOx 360, BioSpherix; Lacona, NY, USA) using nitrogen. Humidity and carbon dioxide were kept at a minimum by continuous adsorption with silica gel and soda lime granules. The chamber was opened every 3 days for less than 20 min in order to clean cages and replenish food and water. SuHx rats received weekly intraperitoneal injections of 20 mg/kg SU5416 suspended in SU-vehicle (0.5% carboxymethylcellulose, 0.4% Tween80 and 0.9% benzyl alcohol in isotonic saline) [31].

Doppler TTE was performed before and after three weeks of hypoxia in rats, as described in the next section. The reduction in the PAT/PET ratio indicated the establishment of PH [29,30,59]. Once disease onset had been confirmed, SuHx rats were then re-exposed to normoxia for two additional weeks and randomly assigned into two groups for oral treatment with either vehicle (DMSO; 100 µL) or LASSBio-1824 (50 mg/kg in DMSO). Normoxic controls were orally administered with DMSO (100 µL). The dose of LASSBio-1824 used was chosen considering previous reports, in which similar N-acylhydrazone compounds (1,2) promoted significant improvement of right ventricular dysfunction and vascular remodeling in monocrotaline-induced PH [28,60].

### 4.3. Transthoracic Echocardiography (TTE)

Under spontaneous ventilation, animals were anesthetized initially with 3% and followed by 1.5% of isoflurane/oxygen mixture. Each examination, from anesthesia to acquiring recordings, required not more than 10 min. RV hemodynamics and cardiac structure and function were assessed using a high-resolution ultrasound imaging system equipped with a RMV-710B transducer with 25 MHz and a fixed focal length of 15 mm (Vevo 770, Visualsonics, Toronto, Canada). All measurements were obtained according to the American Society of Echocardiography Guidelines.

Assessment of pulmonary vascular resistance was performed noninvasively through measurement of PAT obtained by pulsed-wave Doppler TTE at the RV outflow tract (RVOT) [29,31,33,34]. Values obtained for PAT were normalized to PET, in order to attenuate the influence of possible heart-rate fluctuations unrelated to PH [30,61].

Pulmonary-artery diameter (PAd) and velocity-time integral (VTI) were measured in the parasternal short-axis plane in mid-systole by 2-dimensional and pulsed-wave Doppler, respectively. RV cardiac output (RVCO) was calculated as RVCO = 0.785 × (PAd)^2^ × VTI × heart rate. RV wall thickness was obtained in M-mode in order to determine an indirect index of hypertrophy [31,61]. RV and LV areas were obtained in B-mode and LV stroke volume (LVSV) was calculated using Simpson’s method. LV cardiac output (LVCO) was calculated using LVCO = LVSV × heart rate [29,31].

### 4.4. Vascular and Intraventricular Hemodynamic Measurements

Animals underwent deep anesthesia with ketamine (80 mg/kg, i.p.) and xylazine (15 mg/kg, i.p.) and were maintained on spontaneous ventilation during the procedures. A PTFE30 catheter was inserted into the carotid artery to measure systemic systolic, diastolic and mean arterial pressures, and the same was introduced to the LV to measure the LV systolic pressure (LVSP). The right jugular vein was used to access the right ventricle using a second PTFE30 catheter to record the RV systolic pressure, end diastolic pressure and contraction and relaxation rates (positive and negative dP/dt). All parameters were measured using Lab Chart software (Version 7.0, ADInstruments, Inc.; Sydney, Australia). Immediately after completion, animals were exsanguinated for tissue collection.

### 4.5. Vascular Reactivity of Pulmonary Artery Rings

Isometric tension measurements from pulmonary artery rings in response to phenylephrine and acetylcholine were recorded as described elsewhere [28]. Briefly, after dissection, arterial rings were immersed in oxygenated Tyrode saline solution. After equilibration for 2 h at 1.5 g resting tension, preparations were exposed to increasing concentrations of phenylephrine, and after attaining maximal contraction, to increasing concentrations of acetylcholine.

### 4.6. Morphometric and Histochemical Analysis

Before fixation, the RV was isolated from the LV and ventricular septum (S) and their wet weights were recorded for calculation of Fulton index of hypertrophy, as the ratio RV/(LV + S) [31]. RV tissues were set in immersion in neutral buffered formalin. Left lungs were inflated with neutral buffered formalin to obtain a smooth surface. 

Tissues were prepared using graded ethanol and xylene and embedded in molten paraffin. Paraffin blocks were randomly separated in order to stain and analyze by an independent observer. Paraffin sections (4 μm) of RV and lungs were stained by hematoxylin-eosin and picro-Sirius red methods and analyzed following previously described protocols [31]. Immunostaining for smooth-muscle actin in lung sections was also performed as described earlier [31].

Lung sections were immunostained for TNF-α, iNOS and p38 after dewaxing, endogenous peroxidase quenching in 3% hydrogen peroxide and antigen retrieval in citrate (TNFα and p38) or Tris/EDTA (iNOS) buffer for 30 min at 98 °C. Nonspecific binding block was performed by incubation with 5% BSA in phosphate-buffered saline (PBS) for 60 min before incubating overnight in primary antibodies at 1:50 dilution in 1% BSA-PBS. After washing in PBS, slides were incubated for 2 h in HRP polymer-conjugated secondary antibody (diluted 1:3 in PBS) followed by staining in 3,3′-diaminobenzidine solution. Hematoxylin was used as a counterstain and protein content was expressed by percentage stained area of the total field area under 1000× magnification using the color threshold tool of Fiji software [62].

RV sections were immunostained for c-fos as described above, except for a permeabilization step in PBS containing 0.05% Tween20 for 30 min before nonspecific blocking in 1% normal goat serum in PBS for further 30 min. Nuclear translocation of c-fos protein in RV was expressed by the percentage of stained cardiomyocyte nuclei under 1000× magnification.

### 4.7. Membrane Preparation and Western Blot

Immunoblotting was performed as previously described [31,32]. RV tissues were harvested, stored in lysis buffer and frozen in liquid nitrogen until homogenization. Cardiac tissues were homogenized in a potter glass homogenizer using lysis buffer (12.5% sucrose, 20 mM Tris-HCl pH 7.4, 1 mM EDTA) containing 1 mM phenylmethanesulfonyl fluoride, 1 mM benzamidine, 1 mM dithiothreitol, and 1 μg/mL of protease inhibitors (pepstatin, chymostatin, aprotinin, leupeptin and antipain). The mixture was centrifuged for 5 min at 1000 × *g* and supernatant was collected and frozen. Total protein concentration in each sample was determined using Coomassie Blue reagent. Proteins (50 μg) were separated in 10% SDS-PAGE gel and transferred to a nitrocellulose membrane using a semi-dry system (Bio-Rad; Hercules, CA, USA). Membranes were blocked using PBS 5% nonfat milk and 0.1% Tween20 and incubated with primary antibodies TNF-α (1:1000), active caspase-3 (1:1000), p-p38 (1:1000), iNOS (1:1000) and GAPDH (1:1000), in PBS. Freshly made antibody solution was used twice. After labeling the membranes with primary and secondary antibodies (1:10,000 in PBS), detection of specific bands was performed by chemiluminescence using ImageQuant (LAS4000, GE Healthcare Life Sciences; Chicago, IL, USA). Images acquired were analyzed in Fiji software [62] and individual protein-band densities from 5 biological replicates were normalized to GAPDH.

### 4.8. Statistical Analysis

Data were expressed as means ± standard errors of the mean (SEM) and were analyzed using GraphPrism software (version 6.0; GraphPad, San Diego, CA, USA). Data normality and homogeneity within groups were confirmed using Kolmogorov–Smirnov and Brown–Forsythe tests, respectively. The experimental groups were compared using one-way analysis of variance (ANOVA) with a significance level of *p* < 0.05, followed by a *post hoc* Tukey test. Pearson correlation was used for correlation analysis between vessel wall thickness, endothelial function and changes in the PAT/PET ratio.

## 5. Conclusions

The novelty of this work is the demonstration of beneficial effects of a new orally active p38 inhibitor in rats with chronic hypoxia plus SU5416-induced PH. Oral treatment with LASSBio-1824 during 14 days reduced PH characteristics including altered blood flow and endothelial function in pulmonary arteries and tissue remodeling in lung vessels and RV. Inhibition of p38 by LASSBio-1824 attenuated tissue inflammation by reducing TNF-α and iNOS expressions and prevented apoptosis in response to stress in RV tissues by inhibiting c-fos and caspase-3 activation. The effects promoted by LASSBio-1824 contributed to lung and RV repair, which reinforce the importance of considering the p38 MAPK as a promising target to treat PH. Thus, the present data contribute to the hypothesis that p38 inhibition is an important approach for treatment of this deleterious cardio-pulmonary disease.

## Figures and Tables

**Figure 1 pharmaceuticals-15-00900-f001:**
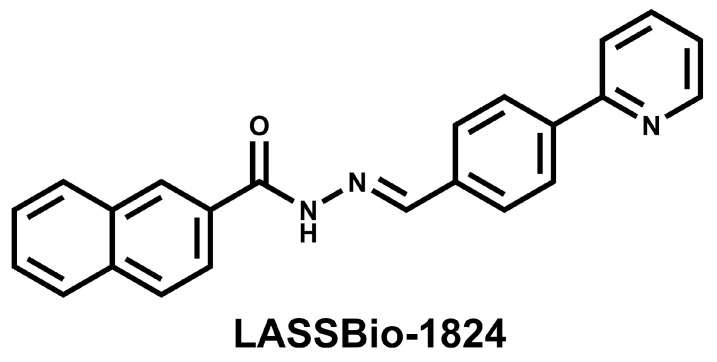
Molecular structure of LASSBio-1824.

**Figure 2 pharmaceuticals-15-00900-f002:**
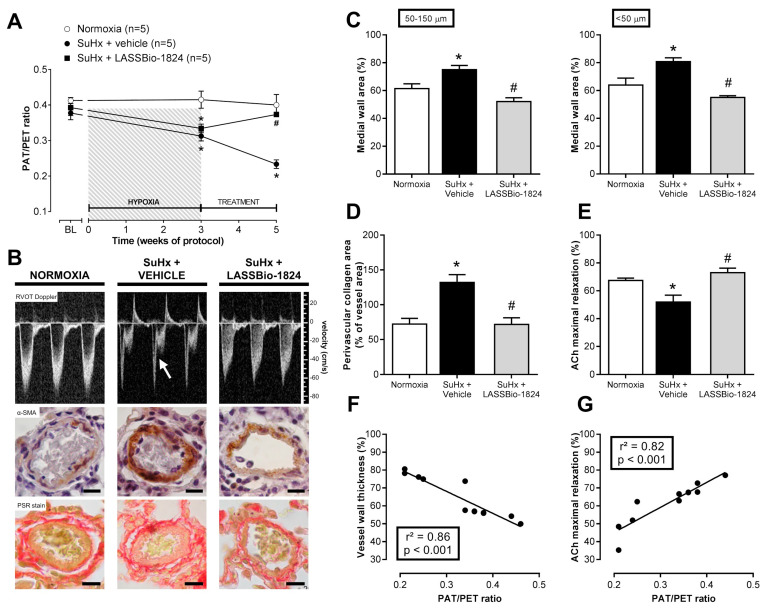
Effect of LASSBio-1824 on pulmonary vascular resistance and remodeling in SuHx. (**A**) Time course of pulmonary vascular resistance development after model induction and treatments (*n* = 5 rats per group). (**B**) Representative images of RVOT Doppler TTE, α-SMA immunostain and PSR stain of lung vessels. Arrow depicts a mid-systolic notch in pulmonary flux profile. Bars represent 20 μm. (**C**), Medial wall hypertrophy of lung vessels (*n* = 5 rats per group). (**D**) Perivascular fibrosis around lung vessels (*n* = 5 rats per group). (**E**) Maximal ACh-induced relaxation of pulmonary arteries (*n* = 5 rats per group). (**F**) Correlation analysis between endpoint pulmonary vascular resistance and wall hypertrophy. (**G**) Correlation analysis between endpoint pulmonary vascular resistance and endothelial function. ACh, acetylcholine; PAT, pulmonary acceleration time; PET, total ejection time; PSR, picro-Sirius red; RVOT, right-ventricle outflow tract; SMA, smooth-muscle actin; TTE, transthoracic echocardiography. * *p* < 0.05 compared to normoxia. ^#^
*p* < 0.05 compared to SuHx + Vehicle.

**Figure 3 pharmaceuticals-15-00900-f003:**
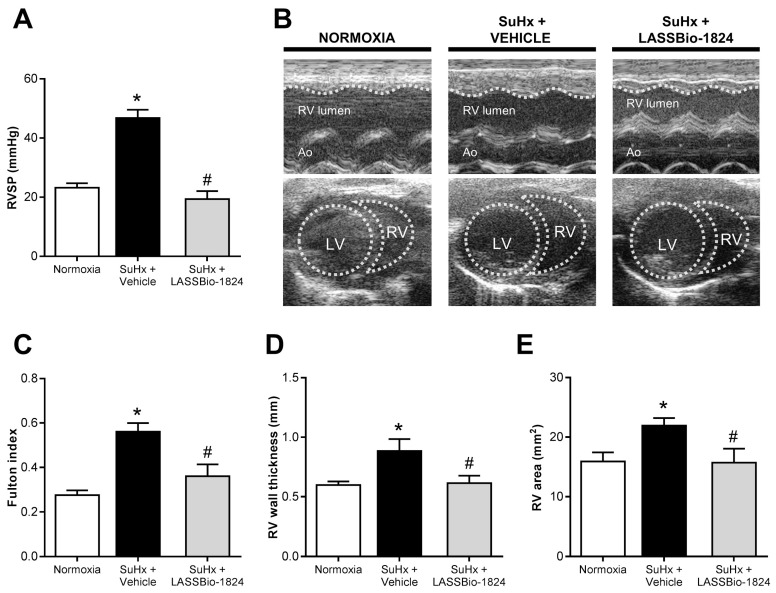
Effect of LASSBio-1824 on right-ventricle remodeling and hemodynamics in SuHx. (**A**) RV systolic pressures after treatment (*n* = 5 rats per group). (**B**) Representative images of cardiac structure by TTE. Traced lines depict ventricular endocardial surfaces. (**C**) Fulton index of hypertrophy of RV (*n* = 5 rats per group). (**D**) Echocardiographic measurement of RV wall hypertrophy (*n* = 5 rats per group). (**E**) Echocardiographic measurement of RV end-diastolic area (*n* = 5 rats per group). Ao, aorta; LV, left ventricle; RV, right ventricle; RVSP, right-ventricle systolic pressure; TTE, transthoracic echocardiography. * *p* < 0.05 compared to normoxia. ^#^
*p* < 0.05 compared to SuHx + Vehicle.

**Figure 4 pharmaceuticals-15-00900-f004:**
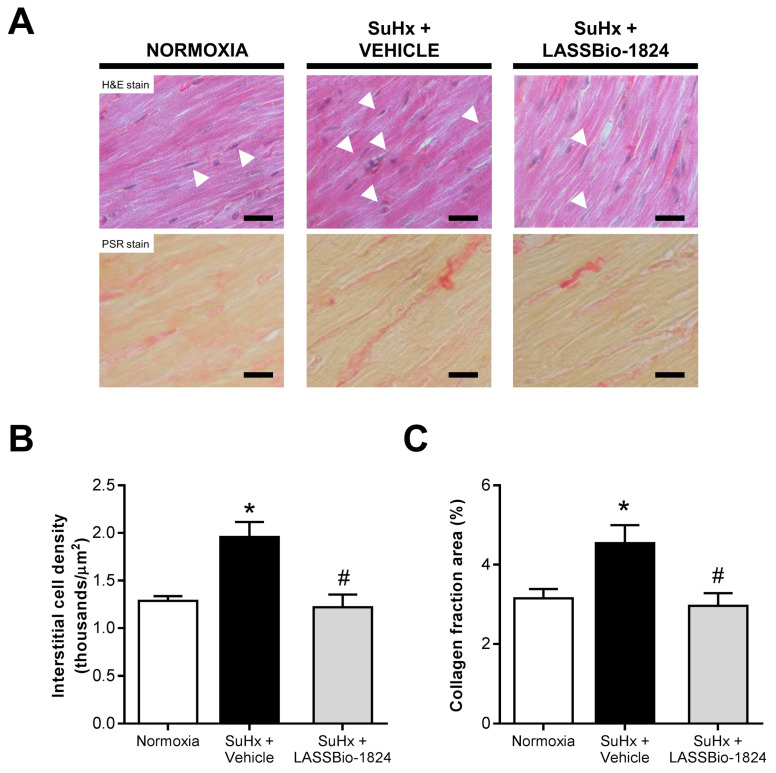
Impact of LASSBio-1824 on RV histology in SuHx. (**A**) Representative micrographs of RV sections with H&E and PSR stains. White arrowheads indicate interstitial cell nuclei. Bars represent 20 μm. (**B**) Interstitial cell density in RV tissue (*n* = 5 rats per group). (**C**) Red-stained collagen area of RV (*n* = 5 rats per group). H&E, hematoxylin-eosin; PSR, picro-Sirius red; RV, right ventricle. * *p* < 0.05 compared to normoxia. ^#^
*p* < 0.05 compared to SuHx + Vehicle.

**Figure 5 pharmaceuticals-15-00900-f005:**
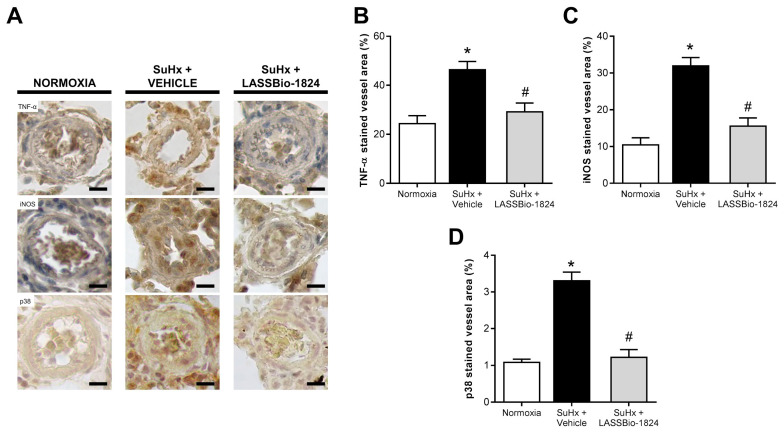
Effect of LASSBio-1824 on lung vasculature content of tissue-inflammation mediators in SuHx. (**A**) Representative micrographs of TNF-α, iNOS and total p38 immunohistochemistry. Bars represent 20 μm. (**B**) Brown-stained wall area of pulmonary vessels corresponding to detected TNF-α (*n* = 5 rats per group). (**C**) Brown-stained wall area of pulmonary vessels corresponding to detected iNOS (*n* = 5 rats per group). (**D**), Brown-stained wall area of pulmonary vessels corresponding to detected total p38 (*n* = 5 rats per group). iNOS, inducible nitric oxide synthase; TNF, tumor necrosis factor. * *p* < 0.05 compared to normoxia. ^#^
*p* < 0.05 compared to SuHx + Vehicle.

**Figure 6 pharmaceuticals-15-00900-f006:**
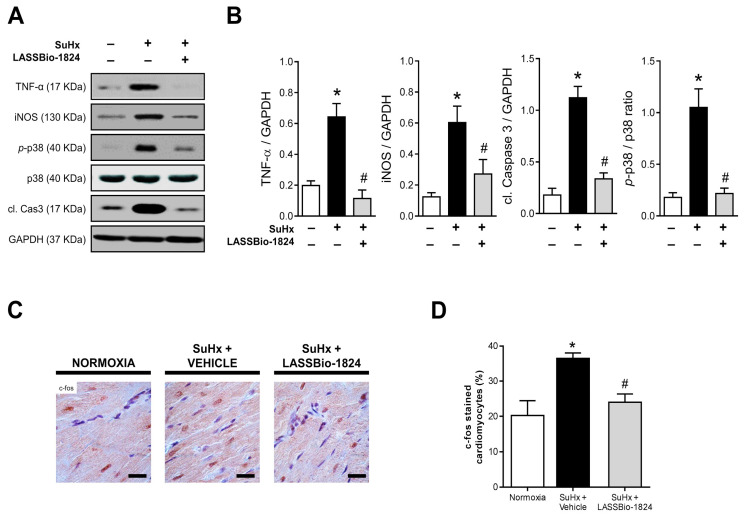
Effect of LASSBio-1824 on RV content of tissue inflammation mediators and cardiac stress and apoptosis in SuHx. (**A**) Representative images of TNF-α, iNOS, phosphorylated and total p38, cleaved caspase 3 and GAPDH immunodetection after Western blot (*n* = 5 biological replicates). (**B**) Relative densities of TNF-α, iNOS, cleaved caspase 3 and phosphorylated p38 in RV samples, respectively (*n* = 5 rats per group). (**C**) Representative micrographs of c-fos immunohistochemistry in RV sections. Bars represent 20 μm. (**D**) Brown-stained cardiomyocyte nuclei abundance in RV samples (*n* = 5 rats per group). cl., cleaved; GAPDH, glyceraldehyde-3-phosphate dehydrogenase; iNOS, inducible nitric oxide synthase; RV, right ventricle; TNF, tumor necrosis factor. * *p* < 0.05 compared to normoxia. ^#^
*p* < 0.05 compared to SuHx + Vehicle.

**Figure 7 pharmaceuticals-15-00900-f007:**
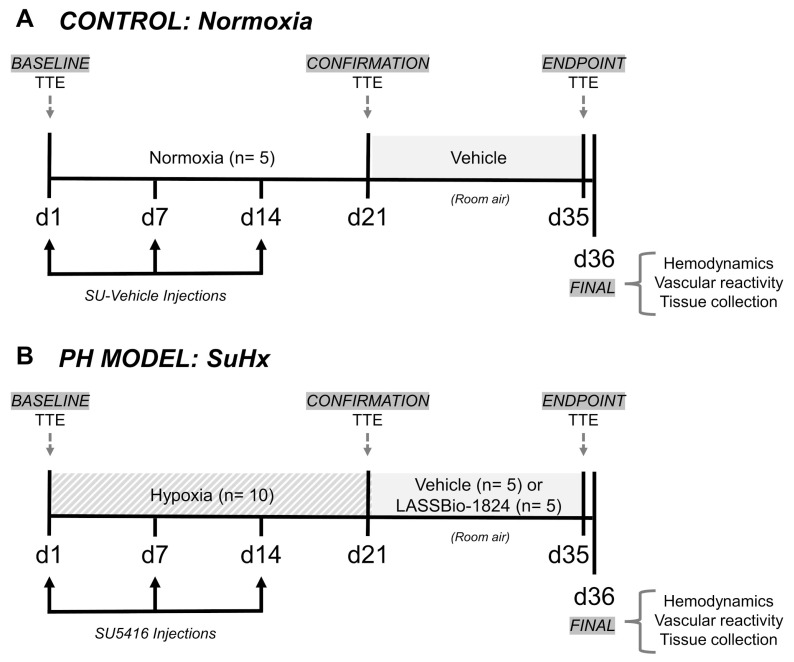
Model induction, therapeutic protocol and measurement schedules. (**A**) Control animals (*n* = 5) were maintained in room air (normoxia, FiO2: 21%) throughout the experimental protocol and received both weekly SU-vehicle (i.p.) and after 3 weeks, daily vehicle (DMSO, 100 μL p.o.). (**B**) SuHx animals were maintained in hypoxic chamber (FiO2: 10%) and received both weekly SU5416 (20 mg/kg i.p.) for 3 weeks and after return to normoxia, daily vehicle (DMSO, 100 μL p.o., *n* = 5) or LASSBio-1824 (50 mg/kg p.o., *n* = 5). TTE, transthoracic echocardiography.

**Table 1 pharmaceuticals-15-00900-t001:** Body morphometry and vascular hemodynamics.

Parameter	Normoxia	SuHx + Vehicle	SuHx + LASSBio-1824
Body weight (g)	236.4 ± 5.5	220.8 ± 9.1	206.5 ± 16.5
Systolic BP (mmHg)	106.9 ± 6.8	104.7 ± 3.8	97.2 ± 8.6
Diastolic BP (mmHg)	79.5 ± 7.1	81.6 ± 4.5	71.3 ± 5.4
Mean BP (mmHg)	93.3 ± 6.9	93.4 ± 4.1	84.2 ± 2.8

Data represent mean ± SEM (*n* = 5 rats per group). BP, blood pressure.

**Table 2 pharmaceuticals-15-00900-t002:** Pulmonary artery dose–response parameters.

Parameter	Normoxia	SuHx + Vehicle	SuHx + LASSBio-1824
Phenylephrine EC_50_ (nM)	62.6 ± 5.9	82.7 ± 22.8	91.1 ± 2.6
Phenylephrine E_max_ (%)	100.0	100.0	100.0
Acetylcholine EC_50_ (nM)	66.1 ± 18.6	72.2 ± 26.2	136.0 ± 49.2
Acetylcholine E_max_ (%)	67.5 ± 2.1	51.9 ± 5.0 *	74.1 ± 3.0 ^#^

Data represent mean ± SEM (*n* = 5 rats per group). * *p* < 0.05 compared to Normoxia. ^#^ *p* < 0.05 compared to SuHx + Vehicle. EC_50_, half maximal effective concentration; E_max_, maximal effect.

**Table 3 pharmaceuticals-15-00900-t003:** Cardiac function and ventricular hemodynamic measurements.

Parameter	Normoxia	SuHx + Vehicle	SuHx + LASSBio-1824
Heart rate (bpm)	283.0 ± 110.2	302.5 ± 25.56	285.1 ± 13.67
LV cardiac output (mL·min^−1^)	86.18 ± 7.55	73.23 ± 3.91	81.36 ± 4.35
LV systolic pressure (mmHg)	98.1 ± 4.1	83.3 ± 5.2	89.6 ± 2.4
LV area (mm^2^)	35.3 ± 3.0	34.2 ± 3.6	32.4 ± 4.3
RV cardiac output (mL·min^−1^)	90.25 ± 3.9	67.45 ± 5.4 *	91.21 ± 6.15 ^#^
RV end diastolic pressure (mmHg)	6.77 ± 1.23	7.01 ± 3.62	4.81 ± 1.66
RV + dP/dt (mmHg·s^−1^)	1162 ± 193	1608 ± 317	1114 ± 209
RV − dP/dt (mmHg·s^−1^)	−1117 ± 171	−1753 ± 426	−1074 ± 165

Data represent mean ± SEM (*n* = 5 rats per group). * *p* < 0.05 compared to Normoxia. ^#^ *p* < 0.05 compared to SuHx + Vehicle. LV, left ventricle; RV, right ventricle.

**Table 4 pharmaceuticals-15-00900-t004:** Antibodies used in this work.

Target	Host	Clonality	Conjugate	Cat. No.	Vendor
p38	Mouse	Monoclonal	–	ab1793	AbCam (Cambridge, MA, USA)
phosphorylated-p38	Rabbit	Polyclonal	–	ab13847	AbCam (Cambridge, MA, USA)
TNF-α	Rabbit	Polyclonal	–	ab7952	AbCam (Cambridge, MA, USA)
iNOS	Rabbit	Polyclonal	–	4511	Cell Signaling (Danvers, MA, USA)
cleaved caspase 3	Rabbit	Polyclonal	–	2982	Cell Signaling (Danvers, MA, USA)
α-SMA	Mouse	Monoclonal	–	5174	Cell Signaling (Danvers, MA, USA)
GAPDH	Mouse	Monoclonal	–	A2547	Sigma-Aldrich (St. Louis, MO, USA)
Rabbit IgG	Goat	Polyclonal	HRP	1706515	Bio-Rad (Hercules, CA, USA)
Mouse IgG	Goat	Polyclonal	HRP	1706516	Bio-Rad (Hercules, CA, USA)
Mouse/Rabbit IgG	Goat	Polyclonal	HRP	414191F	Nichirei Biosciences (Tokyo, Japan)

GAPDH, glyceraldehyde-3-phosphate dehydrogenase; HRP, horseradish peroxidase; IgG, immunoglobulin G; iNOS, inducible nitric oxide synthase; SMA, smooth-muscle actin; TNF, tumor necrosis factor.

## Data Availability

Data is contained within the article.

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
