# Peer review of "Novel p38 Mitogen-Activated Protein Kinase Inhibitor Reverses Hypoxia-Induced Pulmonary Arterial Hypertension in Rats"

_pharmaceuticals, 2022, doi:10.3390/ph15070900_

Round 1

Reviewer 1 Report

The manuscript entitled “Novel p38 Mitogen-Activated Protein Kinase Inhibitor Reverses Hypoxia-Induced Pulmonary Arterial Hypertension In Rats” addresses the beneficial effects of the selective p38 MAPK inhibitor LASSBio-1824  against hypoxia-triggered pulmonary hypertension. Initially, the authors showed that LASSBio-1824 attenuated vascular remodeling, cardiac hypertrophy, and endothelial dysfunction. Then, the authors proceeded to some implicated mechanisms. To this end, the authors demonstrated that these beneficial effects were chiefly driven by dampening cerebral pro-inflammatory signals including TNF-α, iNOS, phosphorylated p38, and c-fos besides downregulating the protein expression of the cleaved form of the pro-apoptotic signal caspase-3.

The manuscript is clearly written, and the current findings are interesting.

Comments:     

1) How did the authors decide about the dose of LASSBio-1824 (50 mg/kg in DMSO) in rats. How is the dose relevant to the human dose using the Human effective dose (HED) formula= animal dose x animal Km/ human Km (Nair AB, Jacob S. A simple practice guide for dose conversion between animals and humans. J Basic Clin Pharm. 2016 Mar;7(2):27-31). Please also provide proper citations for selecting such a dose.

2) In the experimental design, why did not the authors incorporate an additional gp (normoxia + LASSBio-1824 (50 mg/kg p.o., n= 5)). This gp may reveal any potential toxicity of the test agent LASSBio-1824 in rats at the indicated dose?

3) What is the LD50 for LASSBio-1824 in rats? Is the used dose safe?

4) In the discussion section, the authors are advised to explain the potential controversy of the current finding of increased cleaved caspase-3 in SuHx gp which may not be in compliance with previous studies that described resistance to apoptosis of medial cells in pulmonary hypertension.  

In fact, central to the pathobiology of pulmonary hypertension is the process of vascular remodeling. This process involves structural and functional changes to the normal architecture of the walls of pulmonary arteries (PAs) that lead to increased formation of neointima and formation of plexiform lesions. Underlying or contributing to the development of these lesions is hypertrophy, proliferation, migration, and resistance to apoptosis of medial cells (Tajsic and Morrell, 2011, Underlying or contributing to the development of these lesions is hypertrophy, proliferation, migration, and resistance to apoptosis of medial cells, Compr Physiol, Jan;1(1):295-317. doi: 10.1002/cphy.c100026).

5) In the discussion section, the authors are advised to explain how would the p38 MAPK inhibitor normalize vascular resistance as described by the authors in the abstract section?

In fact, the pathogenesis of pulmonary hypertension is multifactorial and involves multiple molecular signaling pathways

So, how would the inhibition of a single pro-inflammatory signal normalize the vascular resistance? Would that implicate that the p38 MAPK pathway is a dominant pathway that controls inflammation? Authors are encouraged to speculate about the answer to the comment and add the answer to the discussion section?

6) In the abstract section, lines 29-31, the authors state that: ”Protein content of TNF-α, iNOS, p38 and caspase-3 were reduced both in lung vessels and RV tissues after treatment and a reduced activation of transcription factor c-fos was found in cardiomyocytes of treated SuHx rats”.

The above statement is not accurate since figure 6 A clearly shows that the total protein content of p38 was not affected. Instead, it was the levels of phosphorylated form of p38 that were lowered by LASSBio-1824. Hence, please rewrite the above statement, for example, to tell that:

”Protein content of TNF-α, iNOS, phosphorylated p38, and caspase-3 were reduced both in lung vessels and RV tissues after treatment and reduced activation of transcription factor c-fos was found in cardiomyocytes of treated SuHx rats”.

Please, revise this issue in the entire manuscript, including lines 268-269 in the discussion section.

7) Likewise, please, revise the statements in the discussion section describing the increased expression of p38 in PAH. For example, in lines 216-218: “Increased p38 expression and activity found in lung vessels from patients with idiopathic PAH further suggest a key role for this MAPK in pulmonary vascular remodeling”.

Was the increase in total p38 or the phosphorylated form of p38 MAPK?

8) In the statistical analysis section, did the authors check data normality and homogeneity before proceeding to one-way ANOVA?

9) In figure legends, the authors are advised to add how many animals in each gp were used to generate the quantified data in the corresponding figure? This also applies to histology and immunohistochemistry

10) In Figure 6, please indicate the number of replicates for these Western blotting target proteins. Were they 3-independent replicates?

11) In all figure legends, the authors are advised to clearly describe what each statistical significance symbol denotes, and the comparison was done versus which gp.

12) More recent references including 2021 and 2022 are advised.

Author Response

Response to reviewers

Editor

(a) We noticed that the article contains western blot experiment. Our editorial office requires the authors to provide the uncropped, untouched, full original images of western blots. Would you please provide full original images of western blot to me via email? Thanks for your cooperation.

Find enclosed in the submission of the revised manuscript, the original images of western blot.

(b) According our rule, the article structure should be as below. Please add conclusions part.

  1. Introduction
  2. Results
  3. Discussion
  4. Materials and Methods
  5. Conclusions

Conclusions have been included in the manuscript.

Page 15, line 487:

“The novelty of this work is the demonstration of beneficial effects of a new orally active p38 inhibitor in rats with chronic hypoxia plus SU5416-induced PH. Oral treatment with LASSBio-1824 during 14 days reduced PH characteristics including altered blood flow and endothelial function in pulmonary arteries and tissue remodeling in lung vessels and RV. Inhibition of p38 by LASSBio-1824 attenuated tissue inflammation by reducing TNF-α and iNOS expressions and prevented apoptosis in response to stress in RV tissues by inhibiting c-fos and caspase-3 activation. The effects promoted by LASSBio-1824 contributed to lung and RV repair, which reinforce the importance of considering the p38 MAPK as a promising target to treat PH. Thus, the present data contribute to the hypothesis that p38 inhibition is an important approach for treatment of this deleterious cardio-pulmonary disease.”

Reviewer 1:

The manuscript entitled “Novel p38 Mitogen-Activated Protein Kinase Inhibitor Reverses Hypoxia-Induced Pulmonary Arterial Hypertension In Rats” addresses the beneficial effects of the selective p38 MAPK inhibitor LASSBio-1824 against hypoxia-triggered pulmonary hypertension. Initially, the authors showed that LASSBio-1824 attenuated vascular remodeling, cardiac hypertrophy, and endothelial dysfunction. Then, the authors proceeded to some implicated mechanisms. To this end, the authors demonstrated that these beneficial effects were chiefly driven by dampening cerebral pro-inflammatory signals including TNF-α, iNOS, phosphorylated p38, and c-fos besides downregulating the protein expression of the cleaved form of the pro-apoptotic signal caspase-3.

 The manuscript is clearly written, and the current findings are interesting.

Comments:    

1) How did the authors decide about the dose of LASSBio-1824 (50 mg/kg in DMSO) in rats. How is the dose relevant to the human dose using the Human effective dose (HED) formula= animal dose x animal Km/ human Km (Nair AB, Jacob S. A simple practice guide for dose conversion between animals and humans. J Basic Clin Pharm. 2016 Mar;7(2):27-31). Please also provide proper citations for selecting such a dose.

The dose of LASSBio-1824 used in this work was selected based on previous reports, in which structurally similar N-acylhydrazone compounds (1,2) promoted significant improvement of right ventricular dysfunction and vascular remodeling, in a different animal model of PH, monocrotaline-induced PH. 

The following phrase was added in Methods section:

Page 13, line 386:

“The dose of LASSBio-1824 used was chosen considering previous reports, in which similar N-acylhydrazone compounds (1,2) promoted significant improvement of right ventricular dysfunction and vascular remodeling in monocrotaline-induced PH.” 

[1] ALENCAR, A. K. N. et al. Beneficial effects of a novel agonist of the adenosine A2A receptor on monocrotaline-induced pulmonary hypertension in rats. British Journal of Pharmacology, v. 169, n. 5, p. 953–962, jul. 2013.

[2] ALENCAR, A. K. N. et al. N-acylhydrazone derivative ameliorates monocrotaline-induced pulmonary hypertension through the modulation of adenosine AA2R activity. International Journal of Cardiology, v. 173, n. 2, p. 154–162, 1 maio 2014.

2) In the experimental design, why did not the authors incorporate an additional gp (normoxia + LASSBio-1824 (50 mg/kg p.o., n= 5). This gp may reveal any potential toxicity of the test agent LASSBio-1824 in rats at the indicated dose?

Despite the unequivocal importance of toxicity studies in drug development, this investigation is performed after the end of proof-of-concept study and demonstration of relevant efficacy. The findings of this work, encourage the next stage of evaluation, considering the potential of the alternative target to treat PH.

3) What is the LD50 for LASSBio-1824 in rats? Is the used dose safe?

Approximately 3 g/kg is the estimated LD50 for LASSBio-1824 when administered orally in rats (ACD Labs Percepta software v. 2012 Release Build 2254). Although the LD50 for LASSBio-1824 was not determined experimentally, for ethical reasons, during the protocol it was not observed alterations in body weight (Table 1), heart rate (Table 3) or behavior using LASSBio-1824 at 50 mg/kg p.o. In addition, given the deteriorated cardiorespiratory function in SuHx, LASSBio-1824 did not aggravate mortality of SuHx animals, which was 15% after 5 weeks and similar to previous reports [1]. Those findings indicated the safety of the dose used in this proof-of-concept study.

[1] BHAT, L. et al. RP5063, a novel, multimodal, serotonin receptor modulator, prevents Sugen 5416-hypoxia-induced pulmonary arterial hypertension in rats. European journal of pharmacology, 2017, v. 810, 83–91.

4) In the discussion section, the authors are advised to explain the potential controversy of the current finding of increased cleaved caspase-3 in SuHx gp which may not be in compliance with previous studies that described resistance to apoptosis of medial cells in pulmonary hypertension. 

In fact, entral to the pathobiology of pulmonary hypertension is the process of vascular remodeling. This process involves structural and functional changes to the normal architecture of the walls of pulmonary arteries (PAs) that lead to increased formation of neointima and formation of plexiform lesions. Underlying or contributing to the development of these lesions is hypertrophy, proliferation, migration, and resistance to apoptosis of medial cells (Tajsic and Morrell, 2011, Underlying or contributing to the development of these lesions is hypertrophy, proliferation, migration, and resistance to apoptosis of medial cells, Compr Physiol, Jan;1(1):295-317. doi: 10.1002/cphy.c100026).

The increased cleaved caspase-3 is observed in right ventricle samples from SuHx animals, which relates to cardiomyocyte apoptosis and progression to a decompensated RV failure [1].

[1] MOSELE, F. et al. Effects of Purple Grape Juice in the Redox-Sensitive Modulation of Right Ventricular Remodeling in a Pulmonary Arterial Hypertension Model. J. Cardiovasc. Pharmacol. 2012, 60 (1), 15–22.

5) In the discussion section, the authors are advised to explain how would the p38 MAPK inhibitor normalize vascular resistance as described by the authors in the abstract section?

In fact, the pathogenesis of pulmonary hypertension is multifactorial and involves multiple molecular signaling pathways

So, how would the inhibition of a single pro-inflammatory signal normalize the vascular resistance? Would that implicate that the p38 MAPK pathway is a dominant pathway that controls inflammation? Authors are encouraged to speculate about the answer to the comment and add the answer to the discussion section?

Further discussion on the relevance of p38 to PH pathophysiology was inserted in ‘Discussion’:

Page 9, line 219:

“PH is a multifactorial disease associated with low survival rate in the absence of adequate treatment [1,3,4]. Mortality is correlated to RV dysfunction, which in turn is a consequence of increased pulmonary vascular resistance and remodeling [1,5,6]. Many cell types and stimuli are involved in the initiation, evolution and stabilization of vascular lesions [6–11]. Inflammatory stimuli in the vessel wall not only induces infiltration of circulating leukocytes but also stimulates endothelial and smooth muscle cells to oppose apoptosis, resulting in a proliferative condition [8,10]. Although these aspects may involve different intracellular pathways, some enzymes such as MAPK are important key factors [12,14–18]. Among MAPK enzymes, the importance of p38 in PH was previously reported, because its expression and activity are increased in patients [12,20,21,26]. The relevance of p38 activation to PH pathogenesis is evidenced by its involvement in different cellular processes triggered by growth factors, inflammatory cytokines, reactive oxygen species, hypoxia and environmental stress [14,15,18]. Signaling through p38 interferes with the balance between cell survival and apoptosis, and mediates inflammatory cytokine production locally [14–18]. Recently, a noncanonical pathway was demonstrated which may also contribute to MAPK signaling perpetuation by a positive feedback of p38 activation [14,15].”

  1. Humbert, M. et al. Pathology and pathobiology of pulmonary hypertension: state of the art and research perspectives. Eur. Respir. J. 2019, 53, 1801887.
  2. Hoeper, M.M. el al. A global view of pulmonary hypertension. Lancet Respir. Med. 2016, 4, 306–322.
  3. Hoeper, M.M.; Ghofrani, H.-A.; Grünig, E.; Klose, H.; Olschewski, H.; Rosenkranz, S. Pulmonary Hypertension. Dtsch. Arztebl. Int. 2017, 114, 73–84.
  4. Wood, C.; Balciunas, M.; Lordan, J.; Mellor, A. Perioperative Management of Pulmonary Hypertension. A Review. J. Crit. Care Med. 2021, 7, 83–96.
  5. Galiè, N.; McLaughlin, V. V.; Rubin, L.J.; Simonneau, G. An overview of the 6th World Symposium on Pulmonary Hypertension. Eur. Respir. J. 2019, 53, 1802148.
  6. Rajagopal, S.; Yu, Y.-R.A. The Pathobiology of Pulmonary Arterial Hypertension. Cardiol. Clin. 2022, 40, 1–12.
  7. Lechartier, B. el al. Phenotypic Diversity of Vascular Smooth Muscle Cells in Pulmonary Arterial Hypertension. Chest 2022, 161, 219–231.
  8. Wang, R. el al. Immunity and inflammation in pulmonary arterial hypertension: From pathophysiology mechanisms to treatment perspective. Pharmacol. Res. 2022, 180, 106238.
  9. Shafiq, M. el al. Involvement of mitogen-activated protein kinase (MAPK)-activated protein kinase 2 (MK2) in endothelial dysfunction associated with pulmonary hypertension. Life Sci. 2021, 286, 120075.
  10. Huertas, A. el al. Chronic inflammation within the vascular wall in pulmonary arterial hypertension: more than a spectator. Cardiovasc. Res. 2020, 116, 885–893.
  11. Huertas, A. et al. Pulmonary vascular endothelium: the orchestra conductor in respiratory diseases. Eur. Respir. J. 2018, 51, 1700745.
  12. Weiss, A. el al. Kinases as potential targets for treatment of pulmonary hypertension and right ventricular dysfunction. Br. J. Pharmacol. 2021, 178, 31–53.
  13. Burton, J.C. el al. Atypical p38 Signaling, Activation, and Implications for Disease. Int. J. Mol. Sci. 2021, 22, 4183.
  14. Canovas, B.; Nebreda, A.R. Diversity and versatility of p38 kinase signalling in health and disease. Nat. Rev. Mol. Cell Biol. 2021, 22, 346–366.
  15. Whitaker, R.H.; Cook, J.G. Stress Relief Techniques: p38 MAPK Determines the Balance of Cell Cycle and Apoptosis Pathways. Biomolecules 2021, 11, 1444, doi:10.3390/biom11101444.
  16. Kim, E.K.; Choi, E.-J. Compromised MAPK signaling in human diseases: an update. Arch. Toxicol. 2015, 89, 867–882, doi:10.1007/s00204-015-1472-2.
  17. Cuadrado, A.; Nebreda, A.R. Mechanisms and functions of p38 MAPK signalling. Biochem. J. 2010, 429, 403–417, doi:10.1042/BJ20100323.
  18. Church, A.C. el al. The reversal of pulmonary vascular remodeling through inhibition of p38 MAPK-alpha: a potential novel anti-inflammatory strategy in pulmonary hypertension. Am. J. Physiol. Cell. Mol. Physiol. 2015, 309, L333–L347, doi:10.1152/ajplung.00038.2015.
  19. Lu, J. et al. Specific inhibition of p38 mitogen-activated protein kinase with FR167653 attenuates vascular proliferation in monocrotaline-induced pulmonary hypertension in rats. J. Thorac. Cardiovasc. Surg. 2004, 128, 850–859.
  20. Mortimer, H.J. et al. p38 MAP kinase: Essential role in hypoxia-mediated human pulmonary artery fibroblast proliferation. Pulm. Pharmacol. Ther. 2007, 20, 718–725.

6) In the abstract section, lines 29-31, the authors state that: ”Protein content of TNF-α, iNOS, p38 and caspase-3 were reduced both in lung vessels and RV tissues after treatment and a reduced activation of transcription factor c-fos was found in cardiomyocytes of treated SuHx rats”.

The above statement is not accurate since figure 6 A clearly shows that the total protein content of p38 was not affected. Instead, it was the levels of phosphorylated form of p38 that were lowered by LASSBio-1824. Hence, please rewrite the above statement, for example, to tell that:

”Protein content of TNF-α, iNOS, phosphorylated p38, and caspase-3 were reduced both in lung vessels and RV tissues after treatment and reduced activation of transcription factor c-fos was found in cardiomyocytes of treated SuHx rats”.

Please, revise this issue in the entire manuscript, including lines 268-269 in the discussion section.

As suggested by reviewer, it has been altered the abstract (lines 29-31). However, the discussion was not altered regarding the lung vessel p38 content (lines 283-295) since it refers to the IHC data obtained with an antibody targeted to total p38, while the RV western blot were done with antibodies targeted to phosphorylated (p-p38) or total p38 to assess the ratio phosphorylated/total.

7) Likewise, please, revise the statements in the discussion section describing the increased expression of p38 in PAH. For example, in lines 216-218: “Increased p38 expression and activity found in lung vessels from patients with idiopathic PAH further suggest a key role for this MAPK in pulmonary vascular remodeling”.

Was the increase in total p38 or the phosphorylated form of p38 MAPK?

Increased total p38 content is observed in lung vessels from patients with PH and this finding was confirmed in this work, which was similar to previous reports in animal models [1]. In contrast, we observed an increase in the activation (phosphorylation) of p38 MAPK in RV samples.

[1] Church, A. C. et al. The Reversal of Pulmonary Vascular Remodeling through Inhibition of P38 MAPK-Alpha: A Potential Novel Anti-Inflammatory Strategy in Pulmonary Hypertension. Am. J. Physiol. Cell. Mol. Physiol. 2015, 309 (4), L333–L347.

8) In the statistical analysis section, did the authors check data normality and homogeneity before proceeding to one-way ANOVA?

Data normality and homogeneity were confirmed using Kolmogorov-Smirnov and Brown-Forsythe tests, respectively. This information was added to statistical analysis section in ‘Methods’ (page 15, line 480).

9) In figure legends, the authors are advised to add how many animals in each gp were used to generate the quantified data in the corresponding figure? This also applies to histology and immunohistochemistry

As suggested by reviewer, legends have been altered.  

10) In Figure 6, please indicate the number of replicates for these Western blotting target proteins. Were they 3-independent replicates?

The number of replicated (5) was indicated in the figure and in ‘Methods’ section.

Page 15, line 474:

“Images acquired were analyzed in Fiji software [62] and individual protein band densities from 5 biological replicates were normalized to GAPDH.”

11) In all figure legends, the authors are advised to clearly describe what each statistical significance symbol denotes, and the comparison was done versus which gp.

All alterations were done as suggested by reviewer.

12) More recent references including 2021 and 2022 are advised.

Updated references were included as suggested.   

Reviewer 2 Report

The manuscript is complete and interesting, congratulations.

Author Response

Authors appreciated the reviewer´s comments.

Reviewer 3 Report

This manuscript is reviewing the effect of LASSBio-1824 on pulmonary hypertension (PH) and right ventricle (RV) failure (GF. Silva: Novel p38 Mitogen-Activated Protein Kinase Inhibitor Reverses Hypoxia-Induced Pulmonary Arterial Hypertension In Rats). The paper contains many of ultrasound and immunostaining data in SuHx rats.

Some minor concerns are:

-         In the part of Introduction, need to interpret briefly the cellular function of the target factors; eg. p38 level in different stages of apoptosis.

-         Several times mentioned in vitro effects but cannot be sure are these cellular or molecular effects eg. line 70.

-         All microscopic images are too small with low quality to see clearly the differences, need to improve them and mark ROIs.

-         Fig. 6A, the Westren blot is not a quantitative method, difficult to intrepret the evaluation of these data.

-         In the figure legends, the type of the applied statistical methods are not specified.

-         Any kind of treatment can be studied easily with simple t-test statistic no need ANOVA and Pearsons.

-         Need to show in the part of Conclusion that what is the relationship between MAPK system and target factors in case of vascular and ventricular failures.

A minor question:

-         Is there any study which interpret the effect of ultrasound within this kind of failures?

Author Response

This manuscript is reviewing the effect of LASSBio-1824 on pulmonary hypertension (PH) and right ventricle (RV) failure (GF. Silva: Novel p38 Mitogen-Activated Protein Kinase Inhibitor Reverses Hypoxia-Induced Pulmonary Arterial Hypertension In Rats). The paper contains many of ultrasound and immunostaining data in SuHx rats.

Some minor concerns are:

-In the part of Introduction, need to interpret briefly the cellular function of the target factors; eg. p38 level in different stages of apoptosis.

The following paragraph has been included in the Introduction:

Page 2, line 63:

“Activation of p38 is implicated in increased production of inflammatory mediators, such as tumor necrosis factor (TNF)-α and inducible nitric oxide synthase (iNOS) [1–3]. In addition, p38 also regulates cell apoptosis mediated by caspases [4–6], which is an important condition involved in vascular remodeling during PH progression.”

  1. Jin, C.; Guo, J.; Qiu, X.; Ma, K.; Xiang, M.; Zhu, X.; Guo, J. IGF-1 induces iNOS expression via the p38 MAPK signal pathway in the anti-apoptotic process in pulmonary artery smooth muscle cells during PAH. J. Recept. Signal Transduct. 2014, 34, 325–331.
  2. Patel, M.; Predescu, D.; Bardita, C.; Chen, J.; Jeganathan, N.; Pritchard, M.; DiBartolo, S.; Machado, R.; Predescu, S. Modulation of Intersectin-1s Lung Expression Induces Obliterative Remodeling and Severe Plexiform Arteriopathy in the Murine Pulmonary Vascular Bed. Am. J. Pathol. 2017, 187, 528–542.
  3. Gong, Y.; Yang, Y.; Wu, Q.; Gao, G.; Liu, Y.; Xiong, Y.; Huang, C.; Wu, S. Activation of LXRα improves cardiac remodeling induced by pulmonary artery hypertension in rats. Sci. Rep. 2017, 7, 6169.
  4. Burton, J.C.; Antoniades, W.; Okalova, J.; Roos, M.M.; Grimsey, N.J. Atypical p38 Signaling, Activation, and Implications for Disease. Int. J. Mol. Sci. 2021, 22, 4183.
  5. Canovas, B.; Nebreda, A.R. Diversity and versatility of p38 kinase signalling in health and disease. Nat. Rev. Mol. Cell Biol. 2021, 22, 346–366.
  6. Whitaker, R.H.; Cook, J.G. Stress Relief Techniques: p38 MAPK Determines the Balance of Cell Cycle and Apoptosis Pathways. Biomolecules 2021, 11, 1444.

-Several times mentioned in vitro effects but cannot be sure are these cellular or molecular effects eg. line 70.

When mentioned the in vitro results, they are related to biochemical/molecular experiments and not cellular effects. To better understanding, the sentence has been altered (l. 73).

“This new N-acylhydrazone derivative inhibited p38 in an in vitro biochemical assay and showed anti-inflammatory activity in vivo after oral administration in mice”

-All microscopic images are too small with low quality to see clearly the differences, need to improve them and mark ROIs.

Images were included in higher resolution as suggested. The ROI used were the full pictures observed in the images, therefore no marks in Figures 2, 4-6 were included.

-Fig. 6A, the Westren blot is not a quantitative method, difficult to intrepret the evaluation of these data.

Protein contents were estimated using semi-quantitative analysis, in which data were normalized to the loading control GAPDH. These findings are in accordance with previous reports on a mouse model of pressure-overload RV failure [1].

[1] Kojonazarov, B. et al. P38 MAPK Inhibition Improves Heart Function in Pressure-Loaded Right Ventricular Hypertrophy. Am. J. Respir. Cell Mol. Biol. 2017, 57 (5), 603–614.

-In the figure legends, the type of the applied statistical methods are not specified.

As suggested by reviewer, the statistical methods were added to the legends.

-Any kind of treatment can be studied easily with simple t-test statistic no need ANOVA and Pearsons.

ANOVA used as a conservative statistical test for simultaneous comparison between more than two means. In order to further demonstrate, a relationship between vascular muscularization and endothelial function considered indicative of vascular resistance, it was used Pearson correlation.

-Need to show in the part of Conclusion that what is the relationship between MAPK system and target factors in case of vascular and ventricular failures.

A paragragh regarding the p38 action on target factors was inserted in ‘Conclusion’ (l. 491):

“Inhibition of p38 by LASSBio-1824 attenuated tissue inflammation by reducing TNF-α and iNOS expressions and prevented apoptosis in response to stress in RV tissues by inhibiting c-fos and caspase-3 activation.”

A minor question:

-Is there any study which interpret the effect of ultrasound within this kind of failures?

To this date, no study has critically reported the effect of ultrasound waves to these failures.